# The Search for Short Baseline Neutrino Oscillation with the ICARUS Detector †

Biswaranjan Behera ‡,§ [ID]

Department of Physics, Colorado State University, Fort Collins, CO 80523, USA; bbehera@fnal.gov
† Presented at the 23rd International Workshop on Neutrinos from Accelerators, Salt Lake City, UT, USA, 30–31 July 2022.
‡ Current address: Department of Physics, University of Florida, Gainesville, FL 32611, USA.
§ For the ICARUS Collaboration.

**Abstract:** The 476-ton active mass ICARUS T-600 Liquid Argon Time Projection Chamber (LArTPC) is a pioneering development that has become the template for neutrino and rare event detectors, including the massive next-generation international Deep Underground Neutrino Experiment. It began operation in 2010 at the underground Gran Sasso National Laboratories and was transported to Fermilab in the US in 2017. To ameliorate the impact of shallow-depth operation at Fermilab, the detector has been enhanced with the addition of a new high granularity light detection system inside the LAr volume along with an external cosmic ray tagging system. Currently in the final stages of commissioning, ICARUS is the largest LArTPC ever to operate in a neutrino beam. On this note, we describe the current status of the ICARUS detector and its achievements in this presentation, and review the plans for ongoing development of the analysis tools needed to fulfill its physics program.

**Keywords:** ICARUS; neutrinos; SBN; PMT; CRT; cosmic rays; LArTPC

## 1. Introduction

Neutrino oscillation is a quantum mechanical phenomenon in which a neutrino created with a specific "lepton flavor" (electron, muon, tau) can later be measured with a different lepton flavor. Despite the well established three-flavour mixing picture within the Standard Model, anomalies at $\Delta m^2 \sim o(eV^2)$ have been observed in the last twenty years suggesting the possible existence of at least a fourth neutrino flavor, named "Sterile Neutrinos". The SBN program at Fermilab has been proposed to provide a definitive clarification. Three detectors, all based on the Liquid Argon TPC technique and placed along the Booster Neutrino Beam (BNB) line, will investigate the possible presence of sterile neutrino states. The SBN program [1] has the unique ability to study appearance and disappearance channels simultaneously when searching for sterile neutrinos at the eV scale. The use of improved simulations and advanced reconstruction methods while combining data from both the near and far detectors will permit the exploration of the current parameter range with a strong $5\sigma$ sensitivity in both appearance and disappearance channels. This achievement is expected within three years of data collection, as shown in Figure 1.

The ICARUS T600 detector [2,3] is made of two identical cryostats (see Figure 2 (left)) filled with about 760 tons of ultra-pure liquid argon. Each cryostat houses two TPCs with a 1.5 m maximum drift path, sharing a common central cathode made of punched stainless steel panels. Charged particles interacting with the liquid argon produce both scintillation light and ionization electrons. Electrons are drifted to the anode by a 500 V/cm electric field made of three parallel wire planes. The electronics are designed to allow continuous read-out, digitization, and independent waveform recording of signals from each wire, allowing for full 3D reconstruction of tracks with a spatial resolution of about 1 mm³. Scintillation light is detected by photomultiplier tubes (PMTs) directly immersed in the liquid argon.

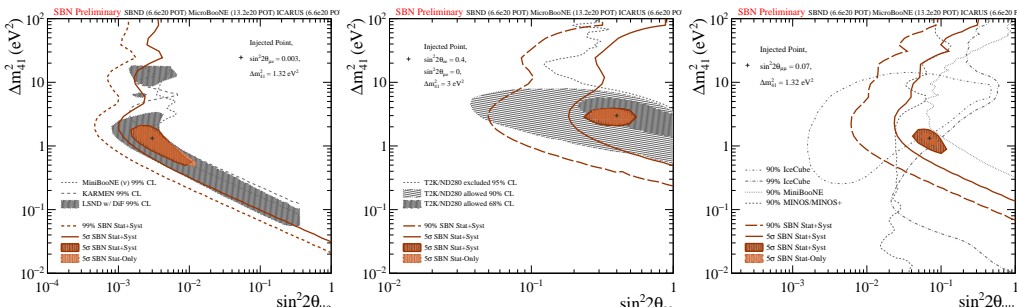

**Figure 1.** The combined analysis of near and far detector data has permitted the currently allowed parameter region to be covered with $5\sigma$ sensitivity in both the appearance and disappearance channels in three years of data taking.

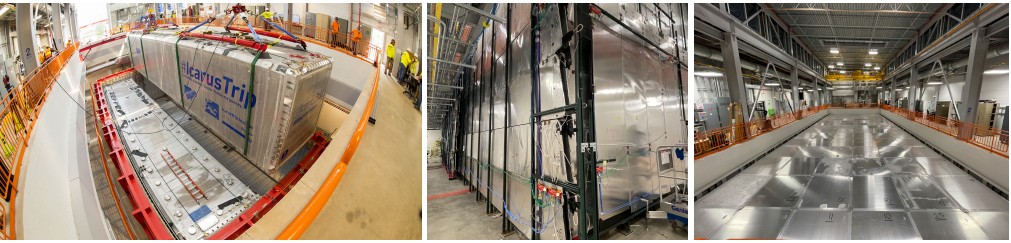

**Figure 2.** The east and west ICARUS cryostats were placed in the SBN Far Detector pit at Fermilab in August 2018 (**left**). A picture of the side CRT is shown (**center**). Installation of the top CRT horizontal modules was completed in December 2021 (**right**). The bottom CRT is not visible in these pictures.

Scintillation light emission in LAr is due to the radiative decay of excimer molecules $Ar^{*2}$ produced by ionizing particles, releasing monochromatic VUV photons ($\lambda \sim 128$ nm) in transitions from the lowest excited molecular state to the dissociative ground state. The emitted light is characterized by fast ($\tau \sim 6$ ns) and slow ($\tau \sim 1.5$ µs) decay components. Their relative intensity depends on dE/dx, ranging from 1:3 for minimum ionizing particles up to 3:1 for alpha particles. These isotropic light signals propagate with negligible attenuation throughout each TPC volume. The light detection system contains 360 Hamamatsu R5912-MOD PMTs installed in groups of ninety devices behind each wire plane of the cryostat. Each PMT is coated with a proper wavelength shifter re-emitting in the visible light range, as the PMT glass windows are not transparent to the scintillation light produced in the liquid argon. The electrical connections between the PMTs and electronics consist of 7 m of cold cables and 37 m of warm cables interconnected by specially designed flanges.

The ICARUS detector is exposed to a huge flux of cosmic rays due to its shallow operating depth at Fermilab. Therefore, cosmic background rejection is important to the achievement of its physics goals. A Cosmic Ray Tagger (CRT) [4] has been constructed surrounding the TPC with a coverage of $4\pi$; it is composed of three subsystems (Figure 2 (center and right)), each with two layers of plastic scintillators and a concrete overburden with 2.85 m thickness placed on the top of the detector. After installation of the overburden, the CRT is less crowded due to fewer hits from low energy particles being received. This allows clear entry and exit points through the CRT to be observed, and results in fewer cosmic ray events, mostly involving muons, inside the TPC. There is a >95% expected tagging efficiency when using only CRT subsystems. Combining the light detection system with the CRT allows for the identification and selection of neutrino and cosmic ray interactions within the BNB and NuMI spill gates.

The Neutrino-4 collaboration [5] has claimed a reactor neutrino disappearance signal that has a clear modulation with L/E$\sim$1–3 m/MeV. ICARUS will be able to test this oscillation hypothesis in the same L/E range in two independent channels with different beams. The disappearance of muon neutrinos from the BNB beam means that analysis can focus on quasi-elastic contained charge current muon neutrino interactions where the muon is at least 50 cm long. In all, $\sim$11,500 events are expected in three months of data taking.

Disappearance of electron neutrinos from the NuMI beam [6] in achieved by selecting the EM shower from quasi-elastic charge current electron neutrino interactions; ~5200 events are expected in a year.

## 2. Commissioning of the ICARUS Detector

After installation [3,7], ICARUS began cooldown and filling in February 2020. In Fall 2020, the detector was activated and the full electric drift field of ~75 kV (500 V/cm) was reached. In order to accurately measure the energy deposition from the ionization charge signal, it is necessary to monitor the purity of the LAr in order to determine the lifetime of free electrons in the LAr. The purity level of the liquid argon is continuously monitored by measuring the signal attenuation in the drift direction along crossing cosmic muon tracks at both the anode and cathode. The electron lifetime reaches up to ~4.5 ms in the East Cryostat and ~3 ms in West Cryostat (Figure 3), allowing efficient signal detection over the full LAr volume.

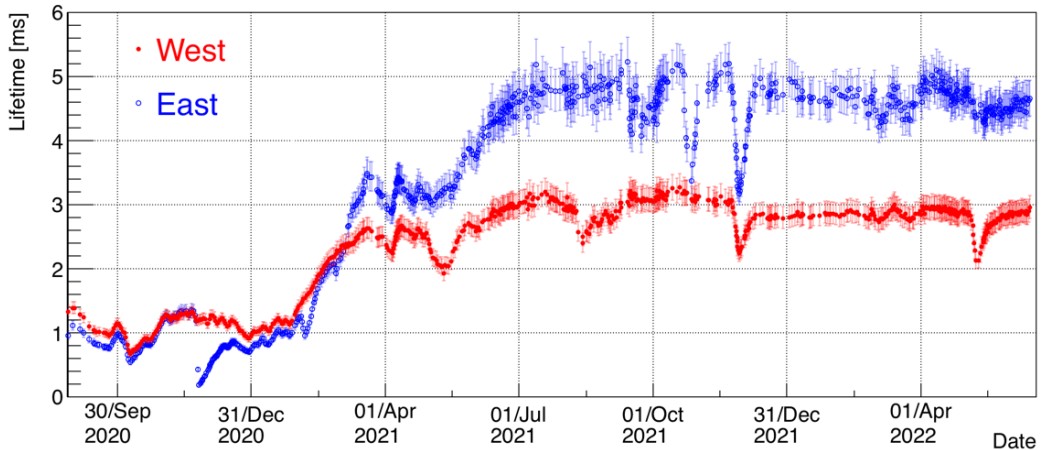

**Figure 3.** Trend of the drift in the electron lifetime in the two ICARUS cryostats during the commissioning phase. The sharp decreases in the lifetime are due to programmed interventions performed on the LAr recirculation pumps or the cryogenic system. It can be seen that the electron lifetime recovers quickly after these interventions.

Commissioning of the TPC started after the TPC wires were biased and soon after the cathode high voltage reached nominal operating conditions. To begin with, noise levels in the TPC were measured using the RMS of the wave from the TPC readout, with an equivalent noise charge of ~550 electrons/ADC. During the commissioning run, a first measurement of the ionization drift velocity in the detector was performed using anode–cathode-crossing cosmic muon tracks as they traversed the full drift length of the detector. The distance between the anode and cathode is 148.2 cm. The time it takes for ionization electrons originating from muon tracks to drift from one end of the track (cathode) to the other (anode) is known as the drift time. The ratio between the distance and drift times provides the drift velocity of the ionization electrons in the liquid argon at the nominal drift electric field of 500 V/cm and temperature of 87.5 K. A correction was made to account for a small bias in precisely reconstructing the drift times associated with the track end points, which was derived from Monte Carlo simulation. A Crystal Ball function was then fit to the maximum ionization drift time distribution associated with cosmic muon tracks in each TPC volume (two per cryostat), with the peak value of each fit used in the ionization drift velocity calculation. The results of the ionization drift velocity measurements in the west cryostat are shown in Figure 4 (left). The results of the measurements, roughly 0.1572 cm/μs for both TPC volumes in the west cryostat, agree with the predicted value of 0.1576 cm/μs to within 0.3% [8,9].

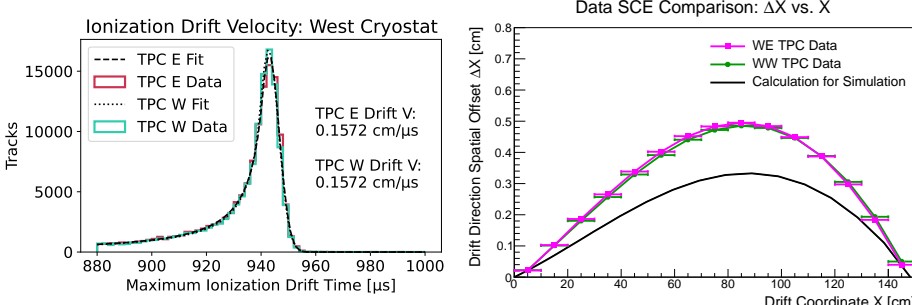

**Figure 4.** Using anode–cathode-crossing cosmic muon ICARUS data, the results show the ionization drift velocity measurement and Crystal Ball fits to the maximum ionization drift time distributions in the two TPCs of the west cryostat (**left**). Using ICARUS commissioning data, the spatial offsets in the drift direction were measured using anode–cathode-crossing cosmic muon tracks as a function of ionization drift distance in the west cryostat (2 TPC) and compared with Monte Carlo simulations of spatial distortions from calculation of the space charge effects (**right**).

In addition to ionization electrons, when charged particles traverse the liquid argon there are slower positively charged argon ions that originate within the detector, a phenomenon known as space charge. Accumulation of space charge causes electric field distortions in near-surface LArTPCs [10–12]. These argon ions drift slowly towards the cathode at a drift velocity of a few millimeters per second under a drift electric field of 500 V/cm [9] and can persist for a long time, creating significant electric field distortions that pull ionization electrons toward the middle of the TPC volume as they drift towards the anode. These distortions lead to bias when reconstructing the point of origin of ionization within the detector, which in LArTPC detectors is referred to as the "spatial distortion". Space charge effects (SCE) can be measured using anode–cathode-crossing cosmic muon tracks, as shown in Figure 4 (right), by looking at spatial distortions in the drift direction. An update of SCEs in ICARUS Monte Carlo simulations using measurements from data is currently in progress.

The trigger system exploits the coincidence of the BNB and NuMI beam spills (1.6 and 9.5 µs, respectively) with the prompt scintillation light produced by charged particles in liquid argon as detected by the PMTs. The generation of the beam spill gates to trigger the readout of the detector is based on receiving "Early Warning" (EW) signals for BNB and NuMI beams 35 and 730 ms, respectively, in advance of the proton on target. Logical signals from the PMT digitizers are processed by programmable FPGA boards to implement a trigger majority logic (a minimum requirement of PMT signals above a threshold) for activation of the ICARUS detector read-out. Additional trigger signals are generated for calibration purposes in correspondence with a subset of the beam spills without any request on the scintillation light (Min-Bias trigger) and outside of the beam spills to detect cosmic ray interactions (Off-Beam trigger) for background modeling. To synchronize all detector subsystems readout with the proton beam spill extraction at the level of a few nanoseconds accuracy, a White Rabbit Network (WR) has been deployed for distributing the beam extraction signals. Absolute GPS timing in the form of pulse per second (PPS) is used as a reference for generating phase-locked digitization clocks (62.5 MHz for the PMT and 10 MHz for the TPC) and for timestamping the beam gates and trigger signals. The trigger system is fully operational. The timing of the beam spills has been initially determined by the difference between the time at which the EW signals arrive and the actual extraction of the proton signal by RWM counters at the target. Figure 5 depicts the neutrino interactions; the accompanying muons of the beam spill in excess of cosmic rays have been clearly identified by requiring at least five fired PMT pairs in the left and right TPCs for both the BNB (left) and NuMI (right) beams. The trigger has performed very well following its completion and commissioning. A visual scanning procedure has been set up to identify neutrino candidate events; in addition, an automatic event selection procedure has been developed and tuned based on visual scanning.

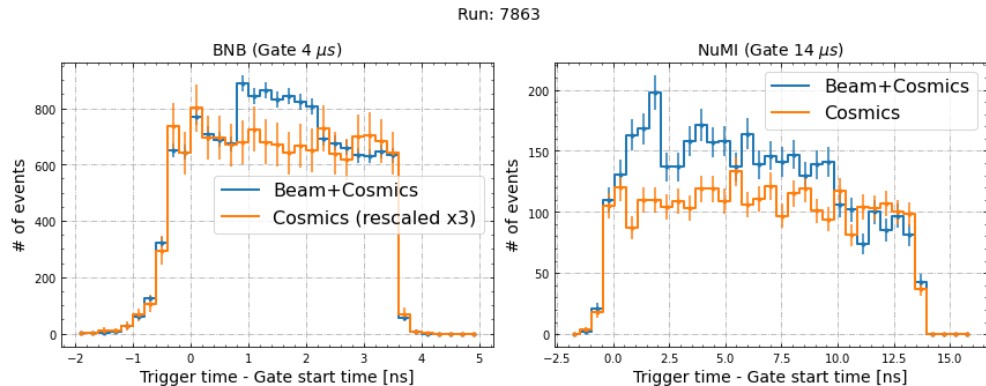

**Figure 5.** Excess of neutrino interactions over cosmic rays in the spill as detected for the BNB (**left**) and NuMI (**right**) beams.

The CRT hit position and time reconstruction algorithm was validated during the commissioning phase [4]. Each CRT hit timestamp was corrected to account for cable delays and light propagation in the scintillator and the wavelength shifter fibre. Figure 6 shows the CRT hit time relative to the neutrino gate start time in the south side CRT wall for the BNB (NuMI) neutrino beam. Using eleven days of commissioning data, a clear peak was observed showing neutrino activity in the 4 μs (12 μs) coincidence window (trigger bias). Additional activity due to the beam appears inside the smaller BNB (NuMI) gate of 1.6 μs within the 4 μs window (9.5 μs within the 12 μs window), with the rest of the activity outside the 1.6 μs (9.5 μs) window being due to cosmic ray triggering. In addition, during commissioning of the overburden it was observed that prior to installation the mean rate was ~610 Hz and 260 Hz for the horizontal and vertical CRT modules, respectively, while after installation these rates were reduced to 330 Hz and 180 Hz, respectively. The overburden [13] is crucial to the ICARUS experiment, as it reduces the cosmogenic background and the data rate.

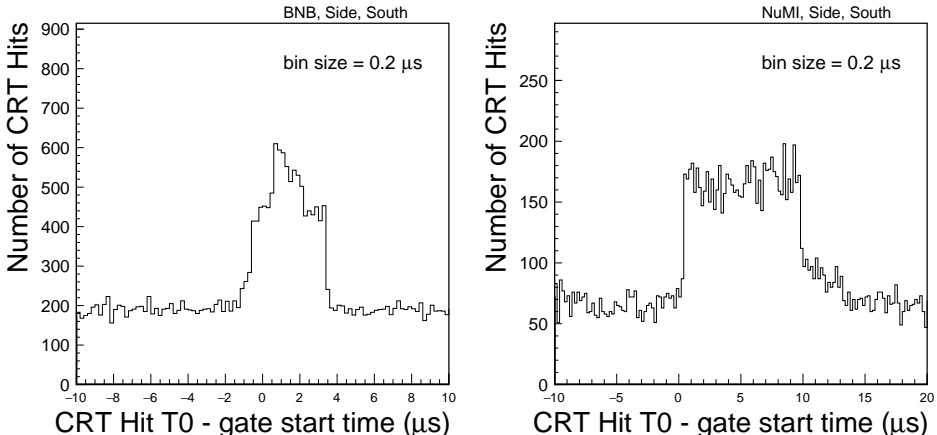

**Figure 6.** CRT hit time relative to the neutrino gate start time in the south wall (side CRT) for the BNB (**left**) and NuMI (**right**) neutrino beams.

### 3. Conclusions and Future Perspectives

The ICARUS detector has operated steadily following its activation on 28 August 2020. After successful commissioning of the detector, a full-time (24/7) neutrino beam physics run was carried out for one month in 2022 before the summer shutdown of the beams. Thus far, using commissioning data with cosmic rays and neutrinos from both the BNB and NuMI has been instrumental in calibrating the detector and tuning simulation and reconstructions tools. The ICARUS detector is well on its way to carrying out intriguing new physics research within the SBN Program.

**Funding:** This research was funded by the US Department of Energy (award DE-SC0017740).

**Institutional Review Board Statement:** Not applicable.

**Informed Consent Statement:** Not applicable.

**Data Availability Statement:** ICARUS Experiment at Fermilab Research Data Policies at: https://icarus-exp.fnal.gov (accessed on 1 August 2022).

**Acknowledgments:** We kindly acknowledge the assistance and encouragement received from ICARUS and SBND collaborators. We appreciate the time and effort on the part of the ICARUS editorial board in carefully reading and providing feedback for this article.

**Conflicts of Interest:** The author declares no conflict of interest.

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
