# Peer review of "The Search for Short Baseline Neutrino Oscillation with the ICARUS Detector†"

_psf, doi:10.3390/psf8010056_

Round 1

Reviewer 1 Report

This manuscript presents a clear and interesting status report of the ICARUS experiment. I find it to be well-written in all its parts. I only have minor comments regarding some references that need to be added.

"The Neutrino-4 collaboration claimed a reactor neutrino disappearance" -> Reference needed (for example: Pisma Zh.Eksp.Teor.Fiz. 109 (2019))

"agree with the predicted value of 0.1576 cm/μs" -> Reference needed

At least a couple of references are necessary both to clarify the effect of the space charge in the TPC and to provide a more solid statement about the possible velocity of the positive ions in argon. This is particularly important for readers who may not be familiar with this subject. For instance:

"there are slow moving positively charged argon ions that originated within the detector, known as space charge. Accumulation of space charge
resulted in electric field distortions in near-surface LAr-TPCs." ->
I recommend adding the reference Astropart.Phys. 92 (2017) 11-20.

"These argon ions, which drift slowly toward the cathode at a drift velocity (a few millimeters per second)" -> Reference needed

In my view, the manuscript can be published with these modifications.

Reviewer 2 Report

Please describe and define the detector and its subsystems more clearly in the introduction section, instead of mentioning one thing at a time when they become relevant to the content. (Such as subsystems in Line 47, East and West Cryostat in Line 70, side CRT, etc.) A figure of the detector would help the readers.

A few of the figures are never mentioned in the main text. Figure 1 needs a lot more explanations. 

Many references are missing, to name a few- ICARUS detector itself, NuMI, Neutrino-4.
